# Estimation of the central 10-degree visual field using en-face images obtained by optical coherence tomography

**Ryu Iikawa[1], Tetsuya Togano[1], Yuta Sakaue[1], Aki Suetake[1], Ryoko Igarashi[1], Daiki Miyamoto[1], Kiyoshi Yaoeda[1,2], Masaaki Seki[1,3], Takeo Fukuchi[1] ***

1 Division of Ophthalmology and Visual Science, Graduate School of Medical and Dental Sciences, Niigata University, Niigata, Japan, 2 Department of Ophthalmology, Yaoeda Eye Clinic, Nagaoka, Niigata, Japan, 3 Department of Ophthalmology, Seki Eye Clinic, Niigata, Japan

* tfuku@med.niigata-u.ac.jp

**Data Availability Statement:** All relevant data are within the manuscript.

**Funding:** The author(s) received no specific funding for this work.

## Abstract

### Purpose

To estimate the central 10-degree visual field of glaucoma patients using en-face images obtained by optical coherence tomography (OCT), and to examine its usefulness.

### Patients and methods

Thirty-eight eyes of 38 patients with primary open angle glaucoma were examined. En-face images were obtained by swept-source OCT (SS-OCT). Nerve fiber bundles (NFBs) on en-face images at points corresponding to Humphrey Field Analyzer (HFA) 10–2 locations were identified with retinal ganglion cell displacement. Estimated visual fields were created based on the presence/absence of NFBs and compared to actual HFA10-2 data. κ coefficients were calculated between probability plots of visual fields and NFBs in en-face images.

### Results

Actual HFA10-2 data and estimated visual fields based on en-face images were well matched: when the test points of <5%, <2%, and <1% of the probability plot in total deviation (TD) and pattern deviation (PD) of HFA were defined as points with visual field defects, the κ coefficients were 0.58, 0.64, and 0.66 in TD, respectively, and 0.68, 0.69, and 0.67 in PD. In eyes with spherical equivalent ≥ −6 diopters, κ coefficients for <5%, <2%, and <1% were 0.58, 0.62, and 0.63 in TD and 0.66, 0.67, and 0.65 in PD, whereas for the myopic group with spherical equivalent < −6 diopters, the values were 0.58, 0.69, and 0.71 in TD and 0.72, 0.71, and 0.71 in PD, respectively. There was no statistically significant difference in κ coefficients between highly myopic eyes and eyes that were not highly myopic.

### Conclusions

NFB defects in en-face images were correlated with HFA10-2 data. Using en-face images obtained by OCT, the central 10-degree visual field was estimated, and a high degree of concordance with actual HFA10-2 data was obtained. This method may be useful for detecting functional abnormalities based on structural abnormalities.

**Competing interests:** The authors have declared that no competing interests exist.

## Introduction

Glaucoma is a condition that involves distinctive changes in the optic nerve and visual field [1,2]. Because glaucoma is a disease characterized by functional and structural abnormalities of the eye, diagnosis and treatment of this condition requires accurate fundus imaging and determination of the visual field. The Humphrey Field Analyzer (HFA; Carl Zeiss Meditec, Dublin, CA, USA) 24–2 or 30–2 program, in which test points are separated by 6 degrees within the central 24 or 30 degrees, is frequently used to evaluate the status of visual function in patients with glaucoma. Examination of the visual field plays a very important role in evaluating vision-related quality of life (VR-QOL) [3,4]. However, an area related to VR-QOL exists in the center of the visual field [5], and the correlation with the National Eye Institute 25-item Visual Function Questionnaire (NEI VFQ-25) score is higher in HFA10-2, in which test points are separated by 2 degrees within the central 10 degrees, than in HFA24-2 [6]. Visual field defects within the central 10 degrees have a large influence on VR-QOL.

Various methods have been developed for observing the retinal nerve fiber layer (RNFL) and nerve fiber layer defects (NFLDs): regular fundus photography, red-free fundus photography [7], scanning laser ophthalmoscopy (SLO) [8], adaptive optics SLO (AO-SLO) [9], scanning laser polarimetry (GDx) [10], Heidelberg Retinal Tomography (HRT) [11], and optical coherence tomography (OCT). In recent years, OCT has been used to detect structural abnormalities; in particular, the circumpapillary retinal nerve fiber layer (cpRNFL) is often used for glaucoma diagnosis. On the other hand, in patients with glaucoma who have central visual field defects in particular, oversights often arise in examinations using the cpRNFL or HFA30-2/24-2 program. Therefore, HFA10-2 is necessary to evaluate VR-QOL [12–16]. However, it is difficult to perform HFA10-2 in all patients due to issues related to factors such as fatigue and measurement time. In addition, because the visual field examination is a psychophysiological test, repeated examinations are necessary, the variation is large, and the result fluctuates in response to patients' psychological factors, physical condition, age, etc.

Hence, in this study we sought to estimate the central 10-degree visual field using the same test points as HFA10-2, based on structural abnormalities in OCT, which takes only a few seconds. Some previous reports have shown a correlation between nerve fiber bundle (NFB) defects in en-face images and visual field defects in HFA10-2 [17–19], but this is the first study to examine the degree of matching for each test point.

## Patients and methods

### Patients

This study was approved by the Ethics Committee of Niigata University (study 2017–0085) and followed the tenets of the Declaration of Helsinki. This is a retrospective study. Information about the research was made public on the Niigata University website, and consent was obtained by giving subjects the opportunity to opt out. We also informed the participants that they have the right to refuse. The ethics committee approved this consent procedure. Thirty-eight eyes of 38 patients with primary open angle glaucoma (POAG) were recruited consecutively, from Niigata University Medical and Dental Hospital between May 2012 and March 2015. If both eyes corresponded to the inclusion criteria in a patient, the eye with worse mean deviation (MD) was enrolled in the study. All glaucoma patients first underwent general ophthalmic examination, including refraction, keratometry, assessment of visual acuity using a Landolt chart, assessment of intraocular pressure using a Goldmann applanation tonometer, gonioscopy, slit-lamp examination, fundus examination, swept-source OCT (SS-OCT) (DRI OCT-1 Atlantis; Topcon, Tokyo, Japan), and visual field examination using the HFA30-2 or

24–2 and HFA10-2 SITA standard strategy. Only single HFA 10–2 data (i.e. a single visit) was used for analysis. Diagnosis of POAG was made according to the guidelines of the Japan Glaucoma Society [2] and the European Glaucoma Society. [20]

Only patients with good reliability in HFA10-2 (fixation loss <25%, false positive <20%, and false negative <33%) within 6 months before or after SS-OCT were included. Patients with a cataract of grade 3 or higher (Emery-Little classification), other types of glaucoma such as primary angle closure glaucoma, pseudoexfoliation glaucoma, steroid-induced glaucoma, uveitic glaucoma, or congenital glaucoma were excluded. Patients with diseases that may affect the sensitivity of the visual field or clarity of en-face images, such as retinal diseases and optic nerve diseases, or with epiretinal membrane (ERM), were also excluded. In order to determine whether similar results could be obtained for highly myopic and eyes that were not highly myopic, we set no exclusion criteria for the spherical equivalent.

## Method

**SS-OCT and acquisition of en-face images.** The examinations of SS-OCT were performed after the pupil was dilated with 0.5% tropicamide and phenylephrine (Mydrin-P; Santen, Osaka, Japan). SS-OCT has an 8-μm depth of resolution in tissue and a 20-μm transverse resolution, and can acquire 100,000 axial scans per second. Imaging was performed with 512×256 axial scans and 6×6-mm cube scans centered at the fovea and optic disc. After flatten processing against the inner limiting membrane (ILM), en-face images were obtained by 6×6-mm 3D scan using the EnView imaging software. At this time, depth from ILM was determined at the point where the nerve fiber bundles (NFBs) were most clearly distinguishable in each case. In order to delineate the defective region and the remaining NFB region in the acquired image, brightness, contrast, and γ value were corrected appropriately. The two images (macula and optic disc) were superimposed using Photoshop (Adobe Systems, San Jose, CA, USA) based on the positions of large blood vessels. Using this method, we were able to observe retinal nerve fibers in the superficial layer at the macula and around the optic disc.

**Creation of estimated visual field.** After the corresponding HFA10-2 locations were identified on the acquired en-face images with retinal ganglion cell (RGC) displacement [21,22], the existence of NFBs at each point was evaluated by three examiners who were blinded to the visual field data. When judgments differed among examiners, the issue was decided by majority vote. The estimated visual field differs by two gradations from a field in which NFB defects are present, i.e., we only estimated whether there was a decline in sensitivity, and could not estimate the degree of visual field sensitivity.

**Comparison with HFA10-2.** Test points with <5%, <2%, and <1% of the probability plot in the total deviation (TD) and pattern deviation (PD) of actual HFA10-2 were defined as points with visual field defects and compared between visual field defects and NFB defects.

We classified subjects into two groups (spherical equivalent ≥ −6 diopters: non-high myopia group and < −6 diopters: high myopia group), and investigated whether there was a difference in the degree of matching. In addition, with reference to Koseki's HFA10-2 cluster [23], we investigated whether there was a difference in the degree of matching between clusters. Clusters are classified as shown in Fig 1; C1 corresponds to the papillomacular area identified in the preliminary study based on the correlation between the foveal threshold and the HFA10-2 examination point, C2 is the upper half of the visual field excluding the papillomacular area, and C3 is the lower half of the visual field excluding the papillomacular area.

**Statistical analysis.** Statistical analysis was performed in Microsoft Excel 2016 using the BellCurve plugin (Social Survey Research Information Co. Japan). The κ coefficient was used to determine the degree of matching. Test points with defects in both the estimated visual field

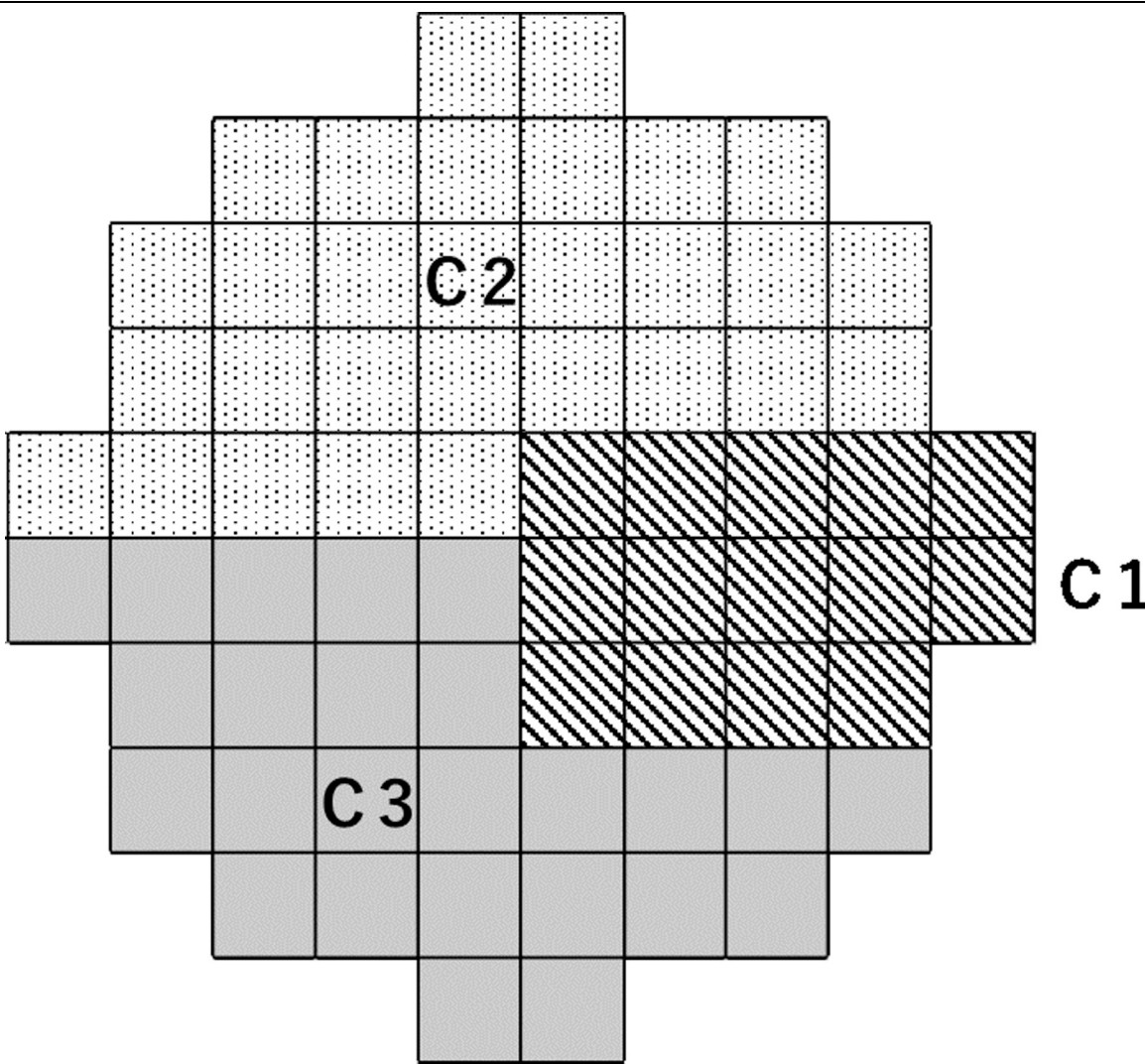

**Fig 1. Cluster classification.** C1 is the papillomacular area, C2 is the upper half of the visual field excluding C1, and C3 is the lower half of the visual field excluding C1.

and actual visual field were defined as "positive," and test points with defects neither in the estimated visual field nor the actual visual field were defined as "negative." Test points with a defect only in the estimated visual field, but not in the actual visual field, were defined as "false positive." Test points with no defect in the estimated visual field, but with a defect in the actual visual field, were defined as "false negative." A total of 68 points were evaluated. Accuracy was determined based on the number of test points with positive and negative judgement i.e. [(positive + negative number)/68] × 100. The κ coefficient was calculated using the positive, negative, false positive, and false negative numbers defined above. The accuracy and κ coefficient were determined for each patient, and the average of 38 patients was calculated. The intraclass correlation coefficient between examiners was determined. The Mann–Whitney U-test was used to compare κ coefficients and accuracy. Statistical significance was defined as $p < 0.05$.

## Results

The profiles of the 38 patients enrolled in this study are shown in Table 1. We recruited and included patients with early- to late-stage glaucoma in this study.

As shown in Fig 2A, an image was obtained by superimposing en-face images of the optic disc and macula areas. After overlaying the points corresponding to HFA10-2 locations on the acquired en-face image with RGC displacement [21,22], the existence of NFBs was determined at each point (Fig 2B). The estimated visual field was created by turning the resultant image upside down (Fig 2C). The actual HFA10-2 TD and PD from a patient are shown in Fig 2D.

Fig 3 shows an example of an en-face image; Fig 3A shows a case in which the papillomacular area was damaged and only a few retinal nerve fiber layers (RNFLs) remained, and Fig 3B shows a case of diffuse RNFL defect.

The average intraclass correlation coefficient between examiners was calculated for each case: the value was 0.94 ± 0.07.

When the test points of <5%, <2%, and <1% of the probability plot in TD and PD were defined as visual field defects, the κ coefficient at all 68 test points was 0.58, 0.64, and 0.66 in TD, respectively, and 0.68, 0.69, and 0.67 in PD (Table 2). There were no significant differences between TD and PD, in any of the probability plots. The accuracy (%) was, respectively, 82.6, 86.8, and 88.3 in TD and 87.7, 88.4, and 88.0 in PD. There were no significant differences between TD and PD, in any of the probability plots. In the group with spherical equivalent ≥ −6 diopters, the κ coefficients for <5%, <2%, and <1% were, respectively, 0.58, 0.62, and 0.63 in TD and 0.66, 0.67, and 0.65 in PD, whereas for the myopic group with spherical equivalent < −6 diopters, the values were 0.58, 0.69, and 0.71 in TD and 0.72, 0.71, and 0.71 in PD (Table 2). There was no difference between the two groups in the degree of matching. In the cluster analysis, the accuracy (%) in TD and PD for the <1% probability plot was 94.4 and 94.4 at C1, 84.5 and 83.3 at C2, and 89.5 and 90.3 at C3, respectively (Table 3). Accuracy was highest in the papillomacular area, and there was a tendency for the accuracy to be higher in the lower visual field than in the upper visual field.

## Discussion

In this report, we estimated the central 10-degree visual field based on the presence or absence of NFBs in en-face images of SS-OCT. This method achieved a high concordance with the results of actual HFA10-2 data. OCT parameters such as cpRNFL and macular ganglion cell complex (GCC) are quantitative and objective tests that can be used to evaluate the presence of glaucomatous changes, as they compare patient data against a database of normal volunteers. These methods are advantageous in terms of objectivity, quantitativeness, and reproducibility. However, these parametric analyses may yield inaccurate results in patients who are highly

**Table 1. Profiles of patients enrolled in this study.**

| Patient (eyes/cases) | 38/38 |
|---|---|
| Age (years, range) | 56.5 ± 9.7 (22 to 78) |
| Sex (female/male) | 19/19 |
| Spherical Equivalent (diopter) | −4.3 ± 4.0 (−11.375 to +2.625) |
| BCVA (LogMAR) | −0.049 ± 0.07 (−0.079 to 0.155) |
| BCVA | 20/18 ± 20/140 (20/30 to 20/16) |
| HFA24-2 or 30–2 MD (dB) | −10.8 ± 6.2 (−0.3 to −23.5) |
| VFI (%) | 67.5 ± 21.0 (20 to 98) |
| HFA10-2 MD (dB) | −12.1 ± 7.6 (−1.65 to −29.84) |

Values represent means ± standard deviation.

POAG, primary open angle glaucoma; BCVA, best-corrected visual acuity; LogMAR, logarithm of the minimum angle of resolution; HFA, Humphrey Field Analyzer; VFI, visual field index; MD, mean deviation.

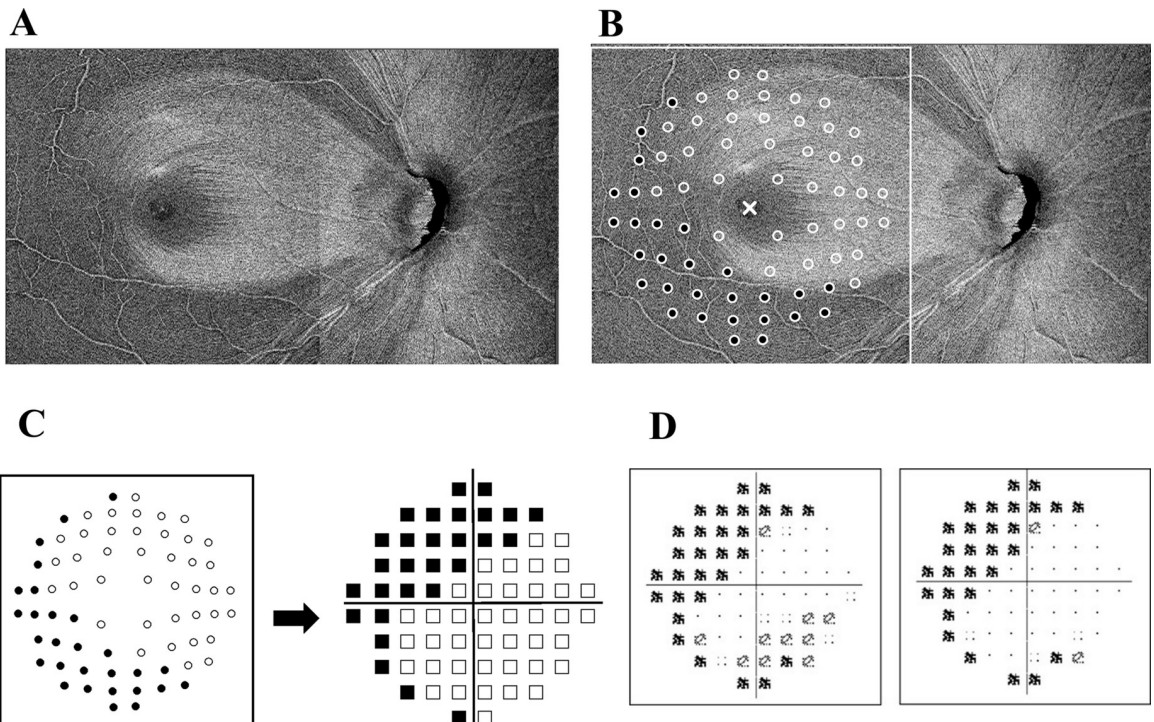

**Fig 2. Creation of estimated visual field.** (A) After superimposition of en-face images of the optic disc and macula area. (B) After points corresponding to HFA10-2 locations were overlaid on the acquired en-face image with RGC displacement, the existence of NFBs was determined at each point. Black circles indicate points where NFBs were absent, and white circles indicate points where NFBs were present. (C) The estimated visual field (right) was created by turning the image at left upside down. (D) TD and PD.

myopic [24]. On the other hand, the en-face image of OCT is qualitative but directly observes the retinal nerve fibers. This method uses the principle that normal RNFL is highly reflective. Because the en-face image does not rely on comparisons to a normal database, it can be adapted to anyone. We grouped the patients into two groups to evaluate whether our methods for estimating the field of view is useful even for patients in whom glaucoma is difficult to diagnose because NFLD may be difficult to evaluate by normal fundus photography due to tigroid fundus, or the use of OCT may be restricted due to severe myopia. As we showed in this study, the accuracy of the estimated visual field did not differ between patients who are highly myopic ($< -6$ diopters) and those who are not highly myopic; therefore, we consider that the en-face image is useful even for patients who are highly myopic.

Several studies have reported the usefulness of en-face images. Jung et al. [25] compared red-free fundus photography with en-face images of OCT and reported that NFLD detected with red-free light was also detectable in en-face images of OCT. Hood et al. [26] described the advantages of en-face images over a RNFL thickness map in OCT, as follows: First, there are few segmentation errors, because segmentation is based on the border between the vitreous and the ILM. Second, blood vessels are easily distinguished from NFB bundles. Third, the RNFL thickness map loses spatial detail due to the segmentation algorithm. Miura et al. [27] reported that the NFLD angle, a new parameter based on the en-face image, has an intra-class correlation coefficient of 0.988, and correlates with the severity of glaucoma (MD and upper and lower TD), making the en-face image useful for glaucoma diagnosis.

We believe that the prominent advantage of the evaluation of the glaucomatous optic disc and retinal changes using en-face images is that RNFL, including the macular area, can be

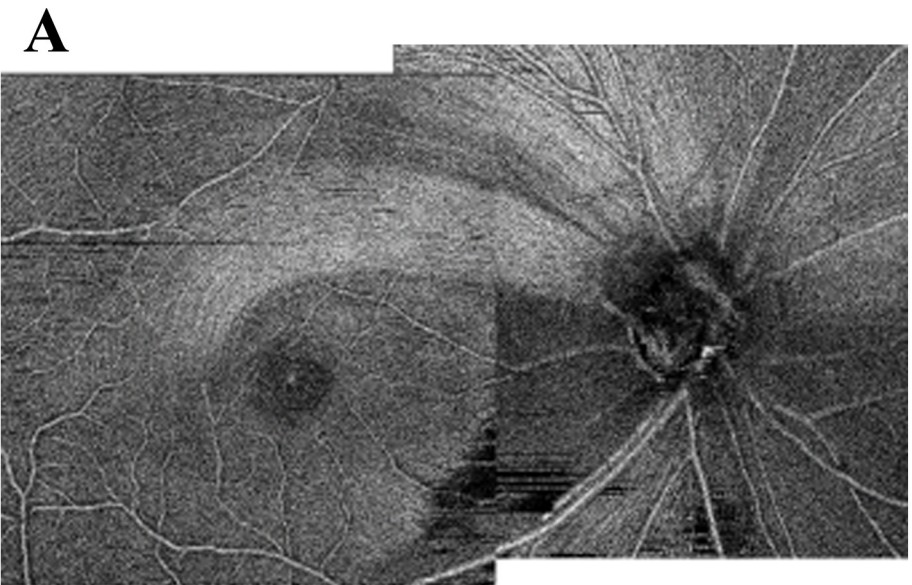

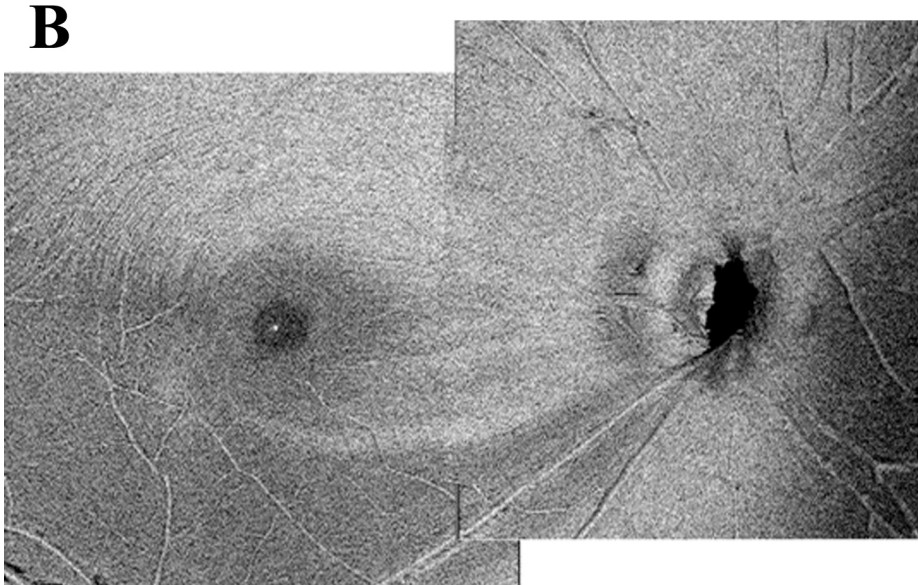

**Fig 3. Example of an en-face image.** (A) A case of damage to the papillomacular area. (B) A case of diffuse RNFL defect.

more clearly visualized. In general examinations such as the visual field test and OCT, emphasis has been placed on how to detect abnormalities; by contrast, we believe that our method is sensitive to detecting remaining RNFL in cases of various stages of glaucoma, especially advanced glaucoma. Moreover, the greatest advantage of the en-face image of OCT is the ability to visualize the presence or absence of residual RNFL in the papillomacular area, and further, to estimate visual field from the status of RNFL. As shown in this study, accuracy was highest in the papillomacular cluster (Table 3), so this is an excellent method for evaluating the

**Table 2. κ coefficient and accuracy (%), total and by spherical equivalent.**

| κ Coefficient (Accuracy %) | TD <1% | TD <2% | TD <5% | PD <1% | PD <2% | PD <5% |
|---|---|---|---|---|---|---|
| Total | 0.66 (88.3) | 0.64 (86.8) | 0.58 (82.6) | 0.67 (88.0) | 0.69 (88.4) | 0.68 (87.7) |
| Spherical Equivalent ≥ –6 Diopters | 0.63 (88.5) | 0.62 (87.3) | 0.58 (84.9) | 0.65 (87.9) | 0.67 (88.5) | 0.66 (87.9) |
| Spherical Equivalent < –6 Diopters | 0.71 (88.0) | 0.69 (86.0) | 0.58 (78.3) | 0.71 (88.1) | 0.71 (88.1) | 0.72 (87.2) |

TD, total deviation; PD, pattern deviation.

visual field corresponding to this fundus area. Asaoka et al. [28] reported that visual acuity correlated with the sensitivity of HFA10-2, and that the correlation coefficient was high in the papillomacular area, which tended to remain until the end stage. Therefore, the papillomacular area is useful for estimation of VR-QOL. In this study, we used RGC displacement [21,22] to overlay the points corresponding to HFA 10–2 test locations onto the acquired en-face image. RGC is displaced in the macular area [29]. Ohkubo et al. [21], who used RGC displacement, reported a correlation between GCC thickness and all 68 test points corresponding to HFA 10–2. They also reported a high correlation, especially at the four central points, and argued that this method is necessary for assessing the structure–function relationship in the macula. The high degree of matching of the two central points included in the papillomacular area may be due to the use of RGC displacement. However, in this study, there were few cases of papillomacular area defects, and there were many cases with upper visual field defects. The reason for the visual field cluster with lowest estimation accuracy in the upper cluster and the highest accuracy in the papillomacular bundle is that in regions where there is an NFB defect, there is a mismatch between the actual visual field at the boundary when a comparison is made for each test point. As a result, fundus and visual-field discrepancy may have occurred. Moreover, because the upper visual field defect is affected by ptosis, it is possible that the measurement accuracy in the visual field is lowered, and that measurement accuracy in the peripheral visual field is generally lower than that of the central visual field, which might affect the result.

There are several reports comparing NFB defects observed in en-face images with actual HFA10-2. Sakamoto et al. [19], using slab images of 50 μm from ILM for advanced glaucoma, reported that the low-reflectivity area in the en-face image corresponds to HFA10-2, and 91.7% of their cases were consistent with TD. Hood et al. [26] also used 52 μm slab images from ILM. They described the limitation of using a fixed-thickness slab. It is possible to overlook early RNFL diffuse loss and deeper retinal information. RNFL has different thicknesses depending on the individual patient and the area of the retina even in a normal eye, and detailed information on RNFL may be sacrificed because the reflection intensity is averaged. Therefore, they argued that the reflection intensity needs to be analyzed and quantified. They also mentioned that the en-face image is not a replacement for OCT, but rather complementary. Alluwimi et al. [17,18] reported that a pattern similar to the defect found in HFA10-2 was observed in the en-face image. Their en-face image was not a slab with a certain thickness, but

**Table 3. Accuracy (%), by cluster.**

| Accuracy (%) | TD <1% | PD <1% |
|---|---|---|
| Cluster 1 | 94.4 | 94.4 |
| Cluster 2 | 84.5 | 83.3 |
| Cluster 3 | 89.5 | 90.3 |

TD, total deviation; PD, pattern deviation.

a cross-section at a certain distance from ILM, similar to our method. The problem with this method is that it is difficult to identify NFB defects at a single distance from the ILM because the RNFL thicknesses of the macular region on the temporal and nasal sides are different. They tried to overcome this by using different distances from ILM for each region. They reported that the width of the NFB defect observed in en-face images increases with increasing depth from ILM, and that the results can vary depending on the depth that is used. All of these reports are limited to myopia with a spherical equivalent up to −6 diopters.

Regarding estimation of the visual field, Takahashi et al. [30] estimated the central 10-degree visual field from macular OCT using the correlation between RNFL thickness and visual field sensitivity in the 68 test points; they reported a high correlation with the observed HFA10-2 visual field, and that their method would be useful for patients in which visual field examinations are difficult. An advantage of our method compared to their method is detection of residual NFB in advanced cases, which may be a floor effect in OCT results (Fig 4. shows an example). They stated that the nasal area, which is susceptible to glaucomatous damage, had a low correlation between RNFL thickness and visual field sensitivity due to the floor effect. In addition, our method may be more useful for highly myopic patients because the previous study excluded subjects with myopia < −8 diopters.

Many studies have reported correlations between retinal inner layer thickness and visual field sensitivity in the macula [21,31]. To date, however, there have been few reports of a relationship between the en-face image of OCT and retinal inner layer thickness or visual field sensitivity. Sakamoto et al. [19] reported a significant difference in sensitivity between the hyperreflective and hyporeflective area of the en-face image. Alluwimi et al. [18] reported that the response to stimuli corresponding to 25 or 28 dB sensitivity in HFA is almost absent in the area where an NFB defect is observed in the en-face image. Because the degree of matching with the visual field was higher in the TD <1% probability plot than in the <5% probability plot, the point of the NFB defect in the en-face image suggests a large decrease in the sensitivity of visual function; however, we are unable to estimate the degree of the decrease. In an attempt at quantification, Gardiner et al. [32] reported that a decrease in RNFL reflection intensity was associated with deterioration in function, which may improve the relationship between structure and function in glaucoma. Ashimatey et al. [33] reported that the reflectivity of the surface RNFL was strongly correlated with cpRNFL thickness. However, reflectance is affected by the surface of the eye, ocular media, the photographing conditions, and the image processing tool; consequently, there are many challenges to quantitation.

In future studies, it will be necessary to further clarify the relationship between retinal sensitivity and areas with NFB defects in the en-face image, as well as the relationship between the foveal threshold and visual acuity in patients whose papillomacular nerve fibers remain. In addition, it is necessary to devise a more accurate estimation method. Specifically, in order to minimize the influence of artifacts on clinicians' judgment, it might be possible to set a reference by binarizing the image and adding information about the thickness of the retinal inner layer of the macula to the en-face image. It may also be necessary to consider whether the degree of matching can be improved by correcting the test point projected on the fundus due to the axial length, as well as overlaying the index using RGC displacement considering the disc inclination.

This study has several limitations. First, this method estimates the locations and extent of the perimetric defect, but cannot estimate its depth. Second, the judgment of NFB defects is subjective. However, high intraclass correlation coefficients were observed in this study, and examiner variation was small. Third, regardless of axial length, the same figure of RGC displacement is overlaid on the acquired en-face image. The size and area of an object projected onto the fundus differ depending on the axial length; however, because this was a retrospective

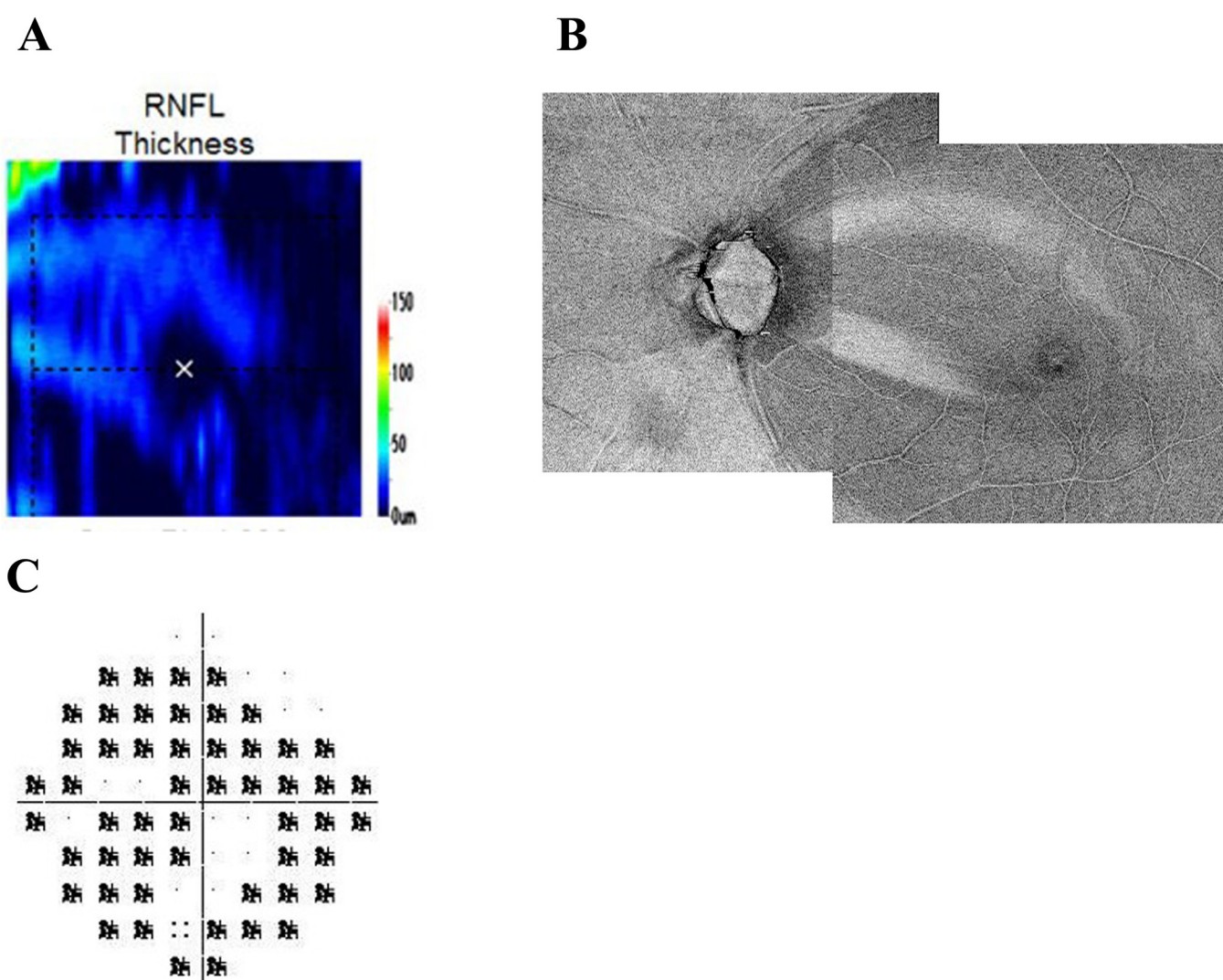

**Fig 4. A case of the floor effect in RNFL thickness.** (A) This case shows the floor effect in RNFL thickness of OCT. (B) Residual NFB is detected more clearly with the en-face image. (C) PD of HFA10-2.

study, data on axial length were insufficient, and this was not corrected. Fourth, we did not consider the inclination of the disc. Fifth, en-face images cannot be obtained in cases with ERM, so our method cannot be applied to all cases. Sixth, there was some bias among the patients in this study: specifically, only a few patients had disorders of the papillomacular area, whereas many more patients had defects in the upper visual field than in the lower visual field. Seventh, because we did not consider the fluctuation and continuity of abnormal points in visual field, the actual visual abnormality of HFA10-2 is not always glaucoma. Hence, multiple averages should be taken. Therefore, it would be desirable to perform additional studies with a larger number of cases and investigate each stage of glaucoma.

In summary, using the OCT en-face image, we could estimate the central 10-degree visual field, and we obtained a high degree of concordance with actual HFA10-2 data. This method may be useful for detecting functional abnormalities based on structural abnormalities.

## Author Contributions

**Conceptualization:** Ryu Iikawa, Takeo Fukuchi.

**Data curation:** Ryu Iikawa, Daiki Miyamoto, Takeo Fukuchi.

**Formal analysis:** Ryu Iikawa, Kiyoshi Yaoeda, Masaaki Seki, Takeo Fukuchi.

**Funding acquisition:** Ryu Iikawa.

**Investigation:** Ryu Iikawa.

**Methodology:** Ryu Iikawa, Takeo Fukuchi.

**Project administration:** Takeo Fukuchi.

**Resources:** Ryu Iikawa, Tetsuya Togano, Yuta Sakaue, Aki Suetake, Ryoko Igarashi, Daiki Miyamoto, Kiyoshi Yaoeda, Masaaki Seki, Takeo Fukuchi.

**Software:** Ryu Iikawa, Daiki Miyamoto.

**Supervision:** Takeo Fukuchi.

**Validation:** Ryu Iikawa, Tetsuya Togano, Yuta Sakaue, Aki Suetake, Ryoko Igarashi.

**Visualization:** Ryu Iikawa.

**Writing – original draft:** Ryu Iikawa.

**Writing – review & editing:** Kiyoshi Yaoeda, Masaaki Seki, Takeo Fukuchi.

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
