## [Decision Letter · Decision Letter 0]

24 Oct 2019

PONE-D-19-21988

Estimation of the central 10-degree visual field using en-face images obtained by optical coherence tomography

PLOS ONE

Dear Dr. Fukuchi,

Thank you for submitting your manuscript to PLOS ONE. After careful consideration, we feel that it has merit but does not fully meet PLOS ONE’s publication criteria as it currently stands. Therefore, we invite you to submit a revised version of the manuscript that addresses the points raised during the review process.

Both learned reviewers have raised serious criticisms that need to be satisfactorily addressed by subjecting the manuscript to a major revision. 

We would appreciate receiving your revised manuscript by Dec 08 2019 11:59PM. To enhance the reproducibility of your results, we recommend that if applicable you deposit your laboratory protocols in protocols.io, where a protocol can be assigned its own identifier (DOI) such that it can be cited independently in the future. For instructions see: http://journals.plos.org/plosone/s/submission-guidelines#loc-laboratory-protocols

We look forward to receiving your revised manuscript.

Kind regards,

Sanjoy Bhattacharya

Academic Editor

PLOS ONE

**Journal Requirements:**

2. Please provide additional details regarding participant consent. In the Methods section, please ensure that you have specified (1) whether consent was informed and (2) what type you obtained (for instance, written or verbal). If your study included minors, state whether you obtained consent from parents or guardians. If the need for consent was waived by the ethics committee, please include this information.

https://iovs.arvojournals.org/article.aspx?articleid=2451212&resultClick=1

In your revision ensure you cite all your sources (including your own works), and quote or rephrase any duplicated text outside the methods section. Further consideration is dependent on these concerns being addressed.

**Comments to the Author**

1. Is the manuscript technically sound, and do the data support the conclusions?

Reviewer #1: Yes

Reviewer #2: Yes

2. Has the statistical analysis been performed appropriately and rigorously? 

Reviewer #1: Yes

Reviewer #2: N/A

3. Have the authors made all data underlying the findings in their manuscript fully available?

Reviewer #1: Yes

Reviewer #2: No

4. Is the manuscript presented in an intelligible fashion and written in standard English?

Reviewer #1: Yes

Reviewer #2: No

5. Review Comments to the Author

Reviewer #1: Major comments:

1. This research is strong, potentially clinically applicable, and original. There are many studies that investigate prediction of glaucomatous visual field progression, but limited research has been done in estimating defects based on nerve fiber layer analysis.

2. The manuscript is technically thorough, and the included data supports the initial research question and conclusions. Statistical analysis was thorough and accurate.

3. Form of consent should be included due to human participants.

4. Discussion: Cases were biased toward more visual field defects in the upper visual field and few in the papillomacular bundle, yet the visual field cluster with lowest estimation accuracy was the upper cluster and the highest accuracy was the papillomacular bundle. The authors should address this more in the discussion, as the upper visual field defect was the most common and least accurate, and the papillomacular bundle defect was the most accurate and the least common. The authors should also address a possible explanation as to why there is a discrepancy between upper and lower visual field defect estimation accuracy.

Minor Comments:

1. Abbreviation RGC should be defined on line #23.

2. Abbreviation NFB should be defined on line #69. It was addressed in the abstract, but this is the first use of the abbreviation in the manuscript.

3. “Goldman” on line #82 should be corrected to Goldmann.

4. The word “defect” on line #305 should be deleted.

Reviewer #2: The purpose of this study was to estimate the perimetric defect within the central degree based on en face images of the RNFL bundles using OCT.

The rational of this study is of a great interest because of the ability to provide more accurate diagnosis of glaucomatous defect, therefore better monitoring of the disease progression within the macula.

However, the estimation was based on the qualitative analysis (en face images of the RNFL bundles), meaning that the prediction would be about locations and extent of the perimetric defect. This concept should be emphasized across the manuscript because one may think that this estimation is about the depth of perimetric defect. Looking at the residual of the RNFL does not necessarily mean there would a certain depth of defect.

Here are some comments as I read through the manuscript:

1. A careful review of the literature is needed. Here are some examples of studies that used en face images with perimetry within the macula:

Hood DC, Fortune B, Mavrommatis MA, et al. Details of Glaucomatous Damage Are Better Seen on Oct En Face Images Than on Oct Retinal Nerve Fiber Layer Thickness Maps. Invest Ophthalmol Vis Sci 2015;56:6208-16.

Alluwimi MS, Swanson WH, Malinovsky VE, King BJ. Customizing Perimetric Locations Based on En Face Images of Retinal Nerve Fiber Bundles with Glaucomatous Damage. Translational Vision Science & Technology 2018;7:5-5.

Alluwimi MS, Swanson WH, Malinovsky VE, King BJ. A Basis for Customising Perimetric Locations within the Macula in Glaucoma. Ophthalmic Physiol Opt 2018;38:164-73.

Other studies regarding Quality of lif in glaucoma:

Goldberg I, Clement CI, Chiang TH, et al. Assessing Quality of Life in Patients with Glaucoma Using the Glaucoma Quality of Life-15 (Gql-15) Questionnaire. J Glaucoma 2009;18:6-12.

Garg A, Hood DC, Pensec N, et al. Macular Damage, as Determined by Structure-Function Staging, Is Associated with Worse Vision-Related Quality of Life in Early Glaucoma. Am J Ophthalmol 2018;194:88-94.

2. Line 49: Please define what HFA 24-2. This will lead the readers to understand the following text.

3. Line 52: References are needed. above are two examples.

4. Line 57: the word "fundus" could be deleted.

5. Line 67: It should be mentioned that ±10° of the visual field.

6. Line 68: What is the justification for the time (few seconds)? Are there references? or do you have data support the time consumption that was mentioned?

7. Line 70: Could you please carefully review the literature (examples are above) and rewrite the sentence to clarify what is different in this study?

8. Line 94: the sentence needs to be rephrased, it is not clear.

9. In the method section, it was not clear how many visits were used to assess the 10-2 test.

10. Line 127: could you clarify more the criterion for defining test points that are less 5%, 2% and 1% of abnormality, for example, 2 consecutive points? with the last consecutive visits?

11. In the statistical analysis, it was very nice explaining how estimated versus actual defective points was categorized. However, it was not clear what the statistical test was used, which was the main question in the study. The statistical analysis used was for k coefficient.

12. Line 169: Why would gray scale needed? Because it was mentioned that the estimation was about whether there was a defect or not. Gray scale is more about the degree of defect which was not estimated.

13. Line 211 to 227: this is about literature review about different imaging modules, which could be briefly mentioned in the introduction. In discussion, this information disrupted the flow of the text.

14. It was frequently stated that different imaging modules do not provide quantitative analysis; en face view does not either.

15. Line 229: rephrase the sentence.

16. Line 231: Why would high myopia affect central visual field? Are there references?

17. Line 266: Are there references supporting your references? did compare displaced versus non-displaced ganglion cell bodies?

18. Line 275: could you please show an example to clarify the idea more for the audience?

19. Line 282 to 284 is not clear. Does talk about the degree of sensitivity?

20. Line 285 and 286: There are studies, as mentioned earlier.

21. Line 291: There were actually published work, here are some examples:

Gardiner SK, Demirel S, Reynaud J, Fortune B. Changes in Retinal Nerve Fiber Layer Reflectance Intensity as a Predictor of Functional Progression in Glaucoma. Invest Ophthalmol Vis Sci 2016;57:1221-7.

Ashimatey BS, King BJ, Burns SA, Swanson WH. Evaluating Glaucomatous Abnormality in Peripapillary Optical Coherence Tomography Enface Visualisation of the Retinal Nerve Fibre Layer Reflectance. Ophthalmic Physiol Opt 2018;38:376-88.

6. PLOS authors have the option to publish the peer review history of their article (what does this mean?). If published, this will include your full peer review and any attached files.

Reviewer #1: Yes: Brian Roberts

Reviewer #2: No

---

## [Author Response · Author response to Decision Letter 0]

12 Dec 2019

We have carefully reviewed the comments and have revised the manuscript accordingly. Our responses are given in a point-by-point manner in "Response to Reviwers"

We look forward to hearing from you regarding our submission.

We would be glad to respond to any further questions and comments that you may have.

---

## [Decision Letter · Decision Letter 1]

18 Feb 2020

Estimation of the central 10-degree visual field using en-face images obtained by optical coherence tomography

PONE-D-19-21988R1

Dear Dr. Fukuchi,

We are pleased to inform you that your manuscript has been judged scientifically suitable for publication and will be formally accepted for publication once it complies with all outstanding technical requirements.

With kind regards,

Sanjoy Bhattacharya

Academic Editor

PLOS ONE

Additional Editor Comments (optional):

Reviewers' comments:

Reviewer's Responses to Questions

**Comments to the Author**

1. If the authors have adequately addressed your comments raised in a previous round of review and you feel that this manuscript is now acceptable for publication, you may indicate that here to bypass the “Comments to the Author” section, enter your conflict of interest statement in the “Confidential to Editor” section, and submit your "Accept" recommendation.

Reviewer #3: All comments have been addressed

2. Is the manuscript technically sound, and do the data support the conclusions?

Reviewer #3: Yes

3. Has the statistical analysis been performed appropriately and rigorously? 

Reviewer #3: Yes

4. Have the authors made all data underlying the findings in their manuscript fully available?

Reviewer #3: Yes

5. Is the manuscript presented in an intelligible fashion and written in standard English?

Reviewer #3: Yes

6. Review Comments to the Author

Reviewer #3: This work is interesting and has application potential. The authors have addressed the comments raised in previous review. I would recommend it for publication.

7. PLOS authors have the option to publish the peer review history of their article (what does this mean?). If published, this will include your full peer review and any attached files.

Reviewer #3: No

---

## [Editor Report · Acceptance letter]

20 Feb 2020

PONE-D-19-21988R1 

Estimation of the central 10-degree visual field using en-face images obtained by optical coherence tomography 

Dear Dr. Fukuchi:

I am pleased to inform you that your manuscript has been deemed suitable for publication in PLOS ONE. Congratulations! Your manuscript is now with our production department. 

With kind regards,

on behalf of

Dr. Sanjoy Bhattacharya 

Academic Editor

PLOS ONE